# P2 Purinergic Signaling in the Distal Lung in Health and Disease

**DOI:** 10.3390/ijms21144973

**Published:** 2020-07-14

**Authors:** Eva Wirsching, Michael Fauler, Giorgio Fois, Manfred Frick

**Affiliations:** Institute of General Physiology, Ulm University, Albert-Einstein-Allee 11, 89081 Ulm, Germany; eva.wirsching@uni-ulm.de (E.W.); michael.fauler@uni-ulm.de (M.F.); giorgio.fois@uni-ulm.de (G.F.)

**Keywords:** P2X receptor, P2Y receptor, ATP, alveolus, lung

## Abstract

The distal lung provides an intricate structure for gas exchange in mammalian lungs. Efficient gas exchange depends on the functional integrity of lung alveoli. The cells in the alveolar tissue serve various functions to maintain alveolar structure, integrity and homeostasis. Alveolar epithelial cells secrete pulmonary surfactant, regulate the alveolar surface liquid (ASL) volume and, together with resident and infiltrating immune cells, provide a powerful host-defense system against a multitude of particles, microbes and toxicants. It is well established that all of these cells express purinergic P2 receptors and that purinergic signaling plays important roles in maintaining alveolar homeostasis. Therefore, it is not surprising that purinergic signaling also contributes to development and progression of severe pathological conditions like pulmonary inflammation, acute lung injury/acute respiratory distress syndrome (ALI/ARDS) and pulmonary fibrosis. Within this review we focus on the role of P2 purinergic signaling in the distal lung in health and disease. We recapitulate the expression of P2 receptors within the cells in the alveoli, the possible sources of ATP (adenosine triphosphate) within alveoli and the contribution of purinergic signaling to regulation of surfactant secretion, ASL volume and composition, as well as immune homeostasis. Finally, we summarize current knowledge of the role for P2 signaling in infectious pneumonia, ALI/ARDS and idiopathic pulmonary fibrosis (IPF).

## 1. Introduction

Alveoli in the distal lung are the functional units for gas exchange within mammalian lungs. In the human lung, some 400 million alveoli provide an extensive surface for efficient gas exchange, whilst the very thin alveolar barrier separating blood and air entails a minimal resistance for diffusion of oxygen and carbon dioxide [1]. This intricate structure constitutes various physiological challenges for maintenance of functional integrity and tissue homeostasis, including biophysical properties at the air-liquid interphase, regulation of local alveolar fluid balance, tissue pressure and lymph flow, perfusion matching and hemostatic control, as well as clearance of inhaled particles, containment of commensal microbiota and defense against pathogenic invaders. All of this is achieved by structural and functional adaptations of the alveolar tissue and the cells within alveolar septae.

Alveolar septae are comprised of two continuous cell layers of epithelium and capillary endothelium forming the thin air-blood barrier, and an interstitial space of variable composition and thickness containing fibroblasts [2,3]. The alveolar epithelium consists of type I (ATI) and type II (ATII) alveolar epithelial cells. Both serve essential roles to maintain alveolar homeostasis [4,5,6,7]. ATI cells cover >95% of the alveolar surface [8] and are mainly renowned for their special morphology, which is perfectly designed for efficient gas exchange between the alveolus and the pulmonary capillaries [9]. Together with ATII cells, ATI cells control transepithelial ion and fluid transport to regulate the volume of the alveolar surface liquid (ASL) and prevent flooding of alveoli. The main function of ATII cells is secretion of pulmonary surfactant [10,11,12]. Surfactant reduces the surface tension at the air-liquid interphase to stabilize alveoli during exhalation and facilitate alveolar distension during lung inflation [13,14]. ATII cells are also considered stem cells within the alveolar epithelium that restore epithelial integrity after injury [15,16]. Fibroblasts secrete components of extracellular matrix to provide elastic and structural properties of the lung parenchyma. Pulmonary endothelial cells (EC) line the blood side of the alveolar barrier and are structurally adapted for maintaining blood flow, gas exchange and fluid balance, as well as the recruitment of immune cells [17].

In addition, the lung contains various immune cells for clearance of inhaled pathogens. Their number and composition vary in healthy and diseased alveoli [18]. Alveolar macrophages (AMs) are the most abundant innate immune cells in the distal lung parenchyma. AMs are loosely attached to the epithelial surface and provide a first-line of defense against pollutants and pathogenic microbes that initiate an innate immune response in the lung [3,19]. AMs are central to orchestrate the initiation and resolution of the immune response in the lung. In addition, AMs perform non-immune, tissue-specific, homeostatic functions, most notably clearance of surfactant [20]. The distal lung also harbors interstitial macrophages [20,21,22], and recent, single-cell RNA-sequencing, studies have revealed great macrophage diversity in healthy, malignant and fibrotic lung tissue [23,24]. Lungs are also a reservoir of neutrophils under a steady-state. In the event of an infection or injury, they are promptly activated and recruited to the alveolar compartment, as well as the airways [25]. Neutrophils migrate to sites of inflammation and act in various ways against pathogens including phagocytosis, neutrophil extracellular traps (NET) formation or degranulation [26]. Besides, many other cell types are recruited to the distal lung under pathological conditions.

All of the cells express purinergic P2 receptors (Table 1). Purinergic signaling plays important roles for maintaining alveolar homeostasis, but is also critically involved in development and progression of severe pathological conditions like pulmonary inflammation, acute lung injury (ALI/ARDS), fibrosis or cancer. In the subsequent sections we aim at summarizing the current state on the role of P2 receptors in the distal lung in health and disease.

## 2. Functional Relevance of P2 Receptor Signaling in the Distal Lung

P2 purinergic signaling is initiated through the binding of extracellular nucleotides to P2 receptors expressed on the surface of target cells. Hence, signaling via P2 receptors can be regulated by the expression of P2 receptors on target cells and/or the availability of nucleotides in the extracellular space.

P2 purinoceptors can be classified into two major families: ionotropic P2X or metabotropic P2Y receptors [55,56,57]. P2X receptors are membrane cation channels formed as homo- or heteromeric trimers from seven P2X receptor subunits (P2X_(1–7)_) [58,59,60]. P2X channels open within milliseconds of ATP binding. The binding of ATP between subunits causes subunits to flex together within the ectodomain and separate in the membrane spanning region so as to open a central channel [58,59,61,62]. P2X channels are essentially non-selective cation channels permeable to small monovalent and divalent cations with preferential gating for Na^+^, K^+^ and Ca^2+^. Receptor activation generally leads to a change in membrane potential, initiating subsequent cellular events. Besides the change of membrane potential, a major physiological mechanism by which activated P2X receptors control cellular functions is elevation in intracellular Ca^2+^ concentration, either directly by Ca^2+^ permeation and/or indirectly by activation of voltage-gated Ca^2+^ channels. The increase in intracellular Ca^2+^ activates a broad range of second messenger systems and signaling cascades, and can trigger manifold short- and long-term cellular events [63,64,65]. P2X receptor activation, conductivity and de-sensitization are also modulated by a variety of compounds including divalent cations, protons, lipids, steroids, ethanol and ivermectin [65,66]. There are a number of excellent reviews that summarize the molecular and functional properties of P2X receptors and their pharmacological targeting in great detail [59,60,65,66,67,68].

P2Y receptors are membrane bound, G-protein-coupled receptors (GPCRs) for extracellular nucleotides [56,69]. Eight P2Y receptor subtypes (P2Y_1_, P2Y_2_, P2Y_4_, P2Y_6_, P2Y_11,_ P2Y_12_, P2Y_13_, and P2Y_14_) are currently recognized [70]. Human P2Y receptor subtypes have, in contrast to P2X receptors, different principal agonists: ATP (P2Y_2_ and P2Y_11_), ADP (P2Y_1_, P2Y_12_, and P2Y_13_), UTP (P2Y_2_ and P2Y_4_), UDP (P2Y_6_ and P2Y_14_), and UDPG (P2Y_14_). ATP may act as an antagonist or partial agonist at several P2Y receptor subtypes, including antagonism at the human P2Y_4_ receptors [71,72]. The eight receptors can also be divided into two subfamilies based on sequence homology and second messengers: five G_q_-coupled P2Y_1_-like (P2Y_1_, P2Y_2_, P2Y_4_, P2Y_6_, and P2Y_11_) and three G_i_-coupled P2Y_12_-like (P2Y_12_–P2Y_14_) receptors [72]. Activation of G_q_-coupled receptors results in stimulation of phospholipase C followed by increases in inositol phosphates and mobilization of Ca^2+^ from intracellular stores. P2Y_11_ receptors couple in addition to G_s_ proteins, followed by increased adenylate cyclase activity. Receptors signaling via G_i_ proteins, in contrast, inhibit adenylate cyclase activity or control ion channel activity [70]. Again, excellent reviews have summarized their molecular, functional and pharmacological properties [68,70,72,73,74,75].

### 2.1. Expression of P2 Receptors in the Distal Lung

Within the lung, expression of a wide range of P2 receptors has been reported on either the mRNA or protein level in epithelial cells [27,28,29,30,31], fibroblasts [39,76], endothelial cells [32,33,34,35,37,38], AMs [40,41,43,44,77,78] and neutrophils [46,47,48,49,50,51,52,53,54,79,80,81] (see Table 1 for details). However, care must be taken when analyzing expression in primary isolated cells as purity of most isolations are <90%. Impurities can account for either underestimation of expression levels or the detection of non-specific expression levels.

P2 receptors are also expressed on many other cell types that are recruited to the lung, in particular under pathological conditions. These cells, mainly of the hematopoietic lineage, do not *per se* represent resident cell populations of the lung and do not likely adopt lung-specific differentiation properties. It is beyond the scope of this review to address them all in detail and therefore we refer to recent reviews [82,83]. Among them are monocytes, T cells, natural killer (NK) cells and regulatory T cells (Treg). Although comprehensive data on P2 receptor expression within these cells are often scarce, it seems that all of these express P2X_7_ receptors [82,84,85,86,87], whilst expression of other P2 receptors may vary between cell types.

Despite the widespread expression of many P2X and P2Y receptor subtypes within the cells of the distal lung, specific and significant physiological functions have only been attributed to few P2 receptor isoforms so far; in particular, P2X_4_, P2X_7_ and P2Y_2_ have been studies in greater detail (Table 2). P2X_4_ and P2Y_2_ receptors expressed on the alveolar epithelium are key regulators for surfactant secretion and also contribute to regulation of the alveolar surface liquid (ASL) volume. Activation of P2X_7_ and P2Y_2_ receptors expressed on immune cells is central for host defense in the alveolus (Figure 1).

### 2.2. ATP in the Alveolus

Activation of P2X and P2Y receptors in the alveolus primarily depends on extracellular ATP. ATP is present in the pulmonary hypophase [6]; however, the estimated concentration under resting conditions is in the low nM range [118], well below the EC_50_ values for P2 receptor activation [59,119,120]. A possible reason for this low concentration is rapid hydrolysis of extracellular ATP in the hypophase by cell surface ectonucleotidases CD39 and CD73 [121,122]. Hence, a tight temporal and spatial coordination between ATP release and ATP demand (i.e., P2 receptor activation) is essential for providing appropriate concentrations of ATP [123]. ATP release within individual alveoli can hence adapt surfactant secretion and fluid transport to local demands. ATP is released from primary ATI, ATII or immortalized alveolar cells in response to increased alveolar distension [4,5,6,124,125], or when coming in close proximity to the air-liquid interphase following a decrease in alveolar hypophase height (i.e., due to increased surface tension forces) [126]. Mechanical ventilation results in inflation-induced ATP release in the *ex-vivo* rat lung, likely as a result of alveolar over distension and stretch of alveolar cells [127]. Consistently, it has been demonstrated *in vitro* and *in vivo* that the stretching of alveolar cells is the most potent stimulus for surfactant secretion, likely via the release of ATP [128,129,130,131], which is released from ATI cells via pannexin hemichannels following an increase in intracellular Ca^2+^ levels, subsequent to activation of purinergic P2X_7_ receptors [5]. We recently collected evidence that mechanical distension of ATI-like cells (hAELVi) [132] results in caveolin-1 dependent activation of mechanosensitive piezo1 channels (unpublished observation) that triggers Ca^2+^-entry and subsequent release of ATP via pannexin hemichannels. ATP is also stored in organelles known as lamellar bodies (LBs) and released upon LB exocytosis and surfactant secretion from ATII cells [123].

The release of purine nucleotides from epithelia is also significantly increased under pathophysiological conditions resulting from chronic lung diseases or following trauma-induced damage of the alveolus [28,133,134]. In line, ATP is released following ventilator-induced lung injury (VILI) upon improperly delivered mechanical ventilation (MV) and injurious overdistension of alveoli [135]. Damaged and necrotic tissues discharge large quantities of ATP. Furthermore, immune cells release ATP via vesicular exocytosis and/or activation of connexin or pannexin hemichannels (this has recently been reviewed in detail [103]). For example, ATP is released in a positive feedback loop from phagocytes via pannexin-1 hemichannels in response to prolonged activation of P2X_7_ receptors. A massive ATP release due to disruption or permeabilization of cell plasma membrane (PM) saturates ATP-hydrolyzing enzymes and leads to sustained increased ATP levels. High levels of extracellular ATP act as a DAMP (danger associated molecular pattern) or “danger signal” [96,134,136]. Alternative mechanisms proposed ATP release following hypotonic swelling observed in a surrogate cell line of type II pneumocytes (A549 cells) [137], or from sympathetic/adrenergic nerves, which terminate close to ATII cells in the lung and utilize ATP as a neuro(co)transmitter [138]. However, their physiological roles *in vivo* need to be determined.

### 2.3. Surfactant Secretion 

Pulmonary surfactant is synthesized in ATII cells, stored in lamellar bodies (LBs) and secreted via regulated exocytosis of LBs [139,140,141] to fulfill its biophysical, as well as immunomodulatory functions at the air-liquid interphase within alveoli [17]. The continuous presence of surfactant is crucial for lung mechanics and survival. Its deficiency causes respiratory failure, most impressively manifested as infant respiratory distress syndrome (IRDS) of newborns with immature lungs [141]. Regulated secretion of pulmonary surfactant is therefore essential for proper lung function and is correlated to elevations of the intracellular Ca^2+^ concentration [10]. The P2 agonist ATP has emerged as the most potent physiological agonist for surfactant secretion amongst a variety of para- or endocrine mediators [112,138,139,142,143,144,145]. ATP triggers LB exocytosis via activation of P2Y_2_ receptor and the subsequent increase in the intracellular Ca^2+^ concentration [10,12,112,142,146]. The ATP-induced Ca^2+^-signal in ATII cells consists of at least 2 phases, an initial, short lasting “peak” followed by a “plateau” phase. The peak is the result of inositol 1,4,5-trisphosphate (IP3)-induced Ca^2+^- release form intracellular stores, whereas the plateau depends on Ca^2+^ entry by a yet undefined mechanism [10,12]. The integrated Ca^2+^ signal defines the amount of LB exocytosis [12,146].

Unlike many readily soluble secretory molecules (e.g., neurotransmitters), surfactant is not immediately released following the fusion of LBs with the plasma membrane. The exocytic fusion pore constitutes a physical barrier for the release of this poorly soluble, lipoprotein-like substance. An increase in intracellular Ca^2+^ at the fusion site is required to expand the fusion pore [94,147,148,149]. This is mediated by activation of P2X_4_ receptors expressed on the membrane of LBs. ATP is stored in LBs and upon exocytosis of LBs and opening of the fusion pore, the low pH within LBs is rapidly neutralized and the luminal ATP activates the P2X_4_ receptors on the membrane of the fused LB [125]. This triggers a transient Ca^2+^-influx (FACE, fusion activated Ca^2+^-entry) at the site of the LB fusion and provides the Ca^2+^ necessary for fusion pore expansion and surfactant release [29,91,92,150]. FACE is short lasting, and P2X_4_ receptors need to be re-sensitized by a protonation/deprotonation cycle depending on receptor internalization and recycling [151]. It has also been suggested that P2X_4_ activation and FACE are part of a positive feedback mechanism, which, due to the increase of Ca^2+^ near the apical PM, stimulates additional LB fusions [10]. Consistently, when cells are stimulated with UTP (instead of ATP), which activates the P2Y_2_ but not the P2X_4_ receptor, the exocytic response is blunted [29]. 

### 2.4. Epithelial Fluid Transport 

Efficient gas exchange and surfactant function depend on regulation of the alveolar surface liquid (ASL) volume and composition [152,153,154]. The ASL is a very thin fluid layer (an average thickness of only 0.2 µm, also termed hypophase) [155], and protects the alveolar surface from desiccation whilst providing a minimal resistance (distance) to gas exchange. Malfunctions in regulation of the ASL volume can cause severe disturbances, such as the formation of alveolar edema [156]. Vectorial ion transport across epithelia drives transepithelial fluid flux. This requires asymmetric or differential distribution of ion transporters and other membrane proteins on the apical and basolateral membrane of polarized epithelial cells. The expression and, in part, localization of several sodium, potassium and anion channels in ATI and ATII cells have been described. Yet, a definite picture of the regulation of transepithelial ion transport to maintain ASL volume under physiological, as well as pathological conditions, is still missing [152,153,154,157,158,159].

Some evidence exists for the contribution of P2 receptor signaling in the regulation of the apical fluid in airways [154,160,161,162], but much less is known about their role in alveolar fluid homeostasis. Several ion channels/transporters known to be affected by luminal P2 receptor activation and/or increases in intracellular Ca^2+^, such as ENaC, CFTR, Ca^2+^-activated Cl^-^ channels (CaCC) or Na^+^/K^+^-ATPase [113,163,164,165,166] are expressed on alveolar epithelial cells [154]. However, direct evidence for the involvement of P2 receptors in maintaining ASL homeostasis is elusive. In general, luminal nucleotides and activation of P2Y_2_ receptors induce Cl^-^, or HCO^3-^ secretion and inhibit ENaC-meditated Na^+^ absorption [113,114] resulting in fluid secretion. Within the alveoli, it is generally accepted that fluid resorption prevails to maintain the relatively thin ASL and impaired Na^+^ reabsorption is associated with the formation of pulmonary edema [152,153]. Hence, P2 receptor-mediated ion channel activation might not play a major role for steady-state fluid transport. It is, however, functionally linked to surfactant secretion. FACE via P2X_4_ drives fluid resorption in response to surfactant secretion. This P2X_4_ receptor-mediated, inward-rectifying cation current on the apical side results in vectorial ion transport across the epithelium, which in turn promotes apical to basolateral fluid transport [93]. The localized alveolar fluid resorption results in temporary thinning of the ASL and promotes contact between surfactant and the air-liquid interphase. This is required for adsorption of newly released surfactant into the air-liquid interface and activation of surfactant [93,167].

### 2.5. Host Defense

The epithelial surfaces of the lungs are in direct contact with the environment and exposed to a multitude of particles, microbes and toxicants. A multilayered physical and chemical innate host-defense system evolved to prevent their entry into lung tissue and the circulation [168]. The complex interplay between resident (e.g. epithelial cells, AMs, dendritic cells) and infiltrating immune cells is regulated by various pro- and anti-inflammatory signaling molecules [169], including extracellular nucleotides [83,170]. In particular, extracellular ATP is an endogenous danger signal to activate immune cells [171,172,173]. Whether activation culminates in inflammation depends on the actual cellular and general composition of the inflammatory microenvironment, receptor expression and state on target cells, the kinetics of nucleotide release, degradation and fate of its products. In general, the purinergic impact on inflammation is highly complex, interwoven with many other signaling cascades and the dynamic and integrative behavior of the system is not well understood. Yet, considerable knowledge has been gained with respect to single pro- and anti-inflammatory processes. While P1 adenosinergic receptors are mainly associated with anti-inflammatory outcomes, P2 receptors predominantly exert pro-inflammatory effects [82,170]. Pro-inflammatory processes affected by P2 receptor stimulation include leukocyte chemotaxis, inflammatory cell maturation or polarization and activation with increased cyto- and chemokine release, ROS (reactive oxygen species) production, hemostasis, cytotoxic and phagocytic activity. Anti-inflammatory effects mediated by P2 receptors are related to apoptotic cell death of immune cells, reduced cytokine release and inhibition of chemotaxis.

So far, P2X_7_ and P2Y_2_ receptors expressed on AMs [40,41,44] neutrophils [46,48,51,53] and infiltrating immune cells have gained the highest attention for host defense in the alveolus. Although it needs to be stated, that data on P2X_7_ receptor expression and function in neutrophils are ambiguous. While many studies found expression of P2X_7_ in neutrophils [48,49,50,51,52,81], others reported an absence of P2X_7_ [45,46,50]. Whether these inconsistencies resulted from differences in research protocols or account for subpopulations of neutrophils remains to be answered.

P2X_7_ receptor is critically involved in the sensing of cell damage when stimulated by extracellular ATP concentrations in the lower mM range, close to intracellular concentrations [101]. Efflux of K^+^ with subsequent depletion of intracellular K^+^ content is thought to provoke NLRP3 inflammasome activation in primed phagocytes [174]. The P2X_7_-dependent Ca^2+^-entry triggers an increased ROS and inflammatory lipid production [175] and activation of TNF-α converting enzyme (TACE/ADAM-17) [176]. NLRP3 inflammasome activation provokes the release of the alarmins IL-1β and IL-18 via non-classical pathways, including exosome secretion and shedding of microvesicles [177]. Other alarmins (e.g., IL-33) are also expressed in alveolar epithelial cells and alveolar macrophages [178], and are released in response to pathogen exposure and cell damage [179]. It has been shown in airways that extracellular ATP triggers release of IL-33 [180], but whether ATP and P2 signaling are involved in IL-33 release in alveoli has yet to be confirmed. Recent results, in LPS-primed macrophages, demonstrate that ATP redirects TNF-α from TACE-dependent cell membrane release to membrane packaging in shed microvesicles [181]. Such macrophage-derived microvesicles are highly pro-inflammatory in lungs [181,182]. How microvesicles are activated and release their cargo at remote target sites is not well understood, but it has been suggested that the P2X_7_ receptor, which is incorporated into microvesicle membranes, is involved [183].

Prolonged stimulation of P2X_7_ receptors on phagocytes provokes pannexin-1 opening, subsequent ATP release [103] and causes apoptotic or pyroptotic cell death. ATP is a “find me” signal for apoptotic cell debris mediated by P2Y_2_ receptors expressed on monocytes [115]. Furthermore, autocrine purinergic signaling promotes chemotaxis of myeloid cells [117,184,185,186,187]. ATP release at the leading edge results in positive feedback through P2Y_2_ receptors [53,116]. Interestingly, Wang et al. demonstrated that LPS might act as a stop signal for neutrophil chemotaxis through autocrine P2X_1_ receptor activation and that P2X_1_ signaling was indispensable for LPS-induced neutrophil degranulation and enhanced phagocytosis [88]. In dendritic cells, functional coupling between P2X_7_ receptor and pannexin-1 is necessary for fast migration [86], whereas T cells rely on P2X_4_ receptor [94].

Inflammatory responses are augmented by platelet activation and thrombus formation [188]. Platelets release large amounts of ADP, inflammatory lipids and cytokines upon activation. ADP promotes thrombus formation by positive feedback through P2Y_1_ and P2Y_12_ receptor signaling. 

Stimulation of P2Y_11_ receptor on immature myeloid dendritic cells leads to partial maturation with increased production of IL-10 and reduced release of IL-12p70, favoring type 2 polarization or tolerance [189]. In line with these findings, plasmacytoid dendritic cells display reduced production of type I interferons upon P2Y receptor stimulation, overall suggesting a negative regulatory role of extracellular nucleotides to contain collateral immunogenic tissue damage [190]. 

T-cell activation is enhanced by autocrine ATP within the immune synapse between naïve T cells and antigen-presenting cells or effector T cells and target cells [191,192,193]. This process involves P2X_1_, P2X_4_, P2X_7_, P2Y_12_ and pannexin-1 receptors for coupled ATP release [82]. Despite this activity amplification within the immune synapse of cytotoxic T lymphocytes, P2X_7_ receptor seems to exert an anti-inflammatory effect through its apoptosis-inducing action on cytotoxic cells in general. This has not only been shown for classical T cells, but also NK and invariant natural killer T cells (iNKT) [82]. P2X_7_ receptor expression in iNKT cells is vitamin A-dependent, such that vitamin A-deficiency led to iNKT overpopulation in mouse lungs and other organs [104]. The pro-apoptotic effect of nucleotides is associated with the expression level of P2X_7_ receptor, being highest in Tregs among T lymphocytes [87]. Tregs play an important role for the maintenance of immune homeostasis. They express high amounts of ectonucleotidases CD39 and CD73, thereby favoring an anti-inflammatory microenvironment of high adenosine and low ATP concentrations. IL-6 exposure increases ATP release by Tregs and fosters T-cell polarization into Th17 cells via P2X_7_ receptor [82].

## 3. The Role of P2 Receptors in Lung Disease

A role for P2 purinergic signaling has been reported for almost all major lung diseases [194]. A better understanding of the impact of P2 receptor signaling on the onset and progression of pulmonary pathologies will eventually help to develop targeted and efficient therapies [195,196]. Within the subsequent sections we will specifically focus on eminent diseases of the distal lung, in particular affecting alveolar function and homeostasis. These include infectious pneumonia, ALI/ARDS and pulmonary fibrosis.

### 3.1. Infectious Pneumonia 

Tissue damage during the course of an infectious disease is due to the direct impact of the pathogen on cellular and tissue homeostasis but is also a consequence of toxic actions originating from the immune response induced to contain the invader. Therefore, the severity of disease pathologies depends on an appropriate trade-off between infectious virulence and immunogenic collateral damage.

With respect to the purinergic system, much research has focused on the role of the P2X_7_ receptor with its pro- and anti-inflammatory impact and this topic has recently been extensively reviewed [77,197]. For example, virus-associated ATP release increases production of type 1 interferons that limit virus replication in infected cells. This defense-mechanism was lost in P2X_7_
*knock-out* (*KO*) mice [102]. Tate et al. demonstrated that NLRP3 inflammasomes play a protective role early during the course of influenza A infection in mice but later turns detrimental due to immunopathogenic effects [198]. In an early study Lee et al. [99] applied replication deficient adenovirus at controlled infectious dosages to *wild type* (*WT*) or P2X_7_
*KO* mice. All animals having received low-dose adenovirus tolerated the infection well, while all wild-type animals in the high-dose group died within five days whereas 30 % in the *KO* groups survived. Increased survival was associated with a reduced inflammatory response due to inhibited inflammasome activation. Similar results were achieved in a model of influenza A infection [100]. Therefore, if P2X_7_ receptor is critically involved in the induction of detrimental hyperinflammation, inhibition of the receptor might be of therapeutic potential to limit tissue damage, whereas P2X_7_-signaling is protective by limiting virus replication. Interestingly, pharmacological inhibition of P2X_7_ receptor ameliorated influenza A pneumonia in mice after inoculation of a lethal dose of virus particles and drug application as early as day 1 post infection [199]. P2X_7_ receptor has also been related to pulmonary tuberculosis [77]. P2X_7_ receptor activity seems to be associated to the ability of infected macrophages to kill mycobacteria, although some controversies arose from the fact that P2X_7_
*KO* mice were either more or less susceptible to mycobacterial infection compared to *WT* mice depending on the virulence of applied mycobacterial strains [77,200,201]. In infections with highly virulent strains, P2X_7_ receptor activation leads to pyroptotic cell death of infected macrophages, causing inflammation and the release of bacteria, thus aggravating the disease process [201]. P2X_7_ receptor single-nucleotide loss-of-function polymorphisms seem to be correlated to an impaired capacity of macrophages to clear the bacterium [77]. Mawatwal et al. reported that calcimycin-mediated stimulation of P2X_7_ receptor improves mycobacterial elimination by enhanced autophagy in a Ca^2+^ and IL-12 dependent manner [96,97]. P2X_7_ receptor has also been described to be required for the clearance of *Streptococcus pneumoniae* [52]. In a recent study, Oluto et al. reported that *Streptococcus pneumoniae* inhibits purinergic signaling by increased internalization of P2Y_2_ receptors in alveolar epithelial cells [202]. This might result in reduced surfactant secretion and subsequent alveolar collapse [95].

### 3.2. ALI/ARDS 

Acute respiratory distress syndrome (ARDS) is a clinically defined syndrome resulting from diffuse acute exudative inflammation in the lung parenchyma [203]. It arises from multiple aetiologies, but many cases present with a common histopathological picture of diffuse alveolar damage, characterized by interstitial and intra-alveolar infiltration of inflammatory cells, hyaline membranes and, depending on the disease phase, hyperplasia of ATII cells and fibrotic organization of intra-alveolar edematous material. This detrimental inflammatory response can be triggered by direct impact to the lung such as in blunt chest trauma, pneumonia or a setting of ischemia-reperfusion injury or indirectly, during the course of systemic inflammatory states like in polytrauma, sepsis or acute pancreatitis. The involvement of the purinergic system in the pathogenesis of ALI/ARDS and VILI is well established. Much about the role of P2 receptors in acute inflammation has already been described in previous sections.

There is compelling evidence from *KO* experiments and pharmacological interferences that P2X_7_ is relevant for the pathogenesis of ARDS [98,105,106,107,108,109,110,111,204,205] (Figure 2). A hallmark of ARDS is excessive invasion and activation of neutrophils (PMN) in the alveolar space. The release of toxic mediators from PMNs causes alveolar barrier breakdown, which is central for developing ARDS. Antagonism of the P2X_7_ receptor with AZ106006120, or *KO* of the receptor, reduced neutrophil infiltration and pro-inflammatory cytokine levels in a mouse model of ALI [98]. Li et al. recently reported that mesenchymal stem cell-derived exosomes ameliorated lung injury in a rat model of blunt chest trauma, and that this was mediated through downregulation of P2X_7_ receptor by microRNA-124-3p, a constituent of these exosomes [110].

In addition, the P2Y_12_ receptor is involved in the pathogenesis of ARDS most likely through its effects on platelet activation and aggregation [206,207]. P2Y_1_ and P2Y_14_ have been described to be relevant for platelet induced-leukocyte migration into LPS-treated lungs [208] but P2Y_1_ was not critically involved in lung injury in a model of LPS-induced peritonitis [209]. Specific blocking of P2X_1_ channels improves transfusion-related acute lung injury in the mouse [89]. P2X_4_ receptor has failed to demonstrate a significant contribution to the final outcome in a mouse model of traumatic lung injury [210]. Dixit et al. reported recently that the non-specific P2-antagonist suramin alleviated systemic and lung inflammation as well as lung injury in mouse models of acute pancreatitis [211]. Further, they observed increased plasma levels of ATP, suggesting that ATP was involved in remote activation of the pulmonary immune system, and indeed i.v. apyrase treatment significantly reduced inflammation and lung injury. Conflicting to this hypothesis is that systemic administration of the P2 agonist ATPγS has been shown to protect against LPS-induced lung injury [212], which might be related to chemotaxis inhibiting effects of systemic ATP on neutrophils [213].

Since patients suffering from ARDS often require ventilatory support, VILI is a complication that eventually aggravates ARDS [136,214]. An early study reported the involvement of P2Y receptors in the pathogenesis of VILI [204]. Hasan et al. suggested that desensitization of P2Y_2_ and P2X_4_ receptors might contribute to the development of VILI [95]. Zheng et al. presented results that in a two-hit mouse model of *Pseudomonas aeruginosa* pneumonia with high-pressure ventilation a specific P2Y_6_ receptor antagonist partially attenuated inflammation and lung injury without interfering with the ability of the immune system to clear the bacteria [215].

### 3.3. Idiopathic Pulmonary Fibrosis 

Idiopathic pulmonary fibrosis (IPF) is a progressive, irreversible and fatal disease. The conceptual model for the pathogenesis of IPF postulates that recurrent micro-injuries to ageing alveolar epithelium result in aberrant epithelial–fibroblast communication, the induction of matrix-producing myofibroblasts, and considerable extracellular matrix accumulation and remodeling of lung interstitium [216,217,218]. Ultimately, this leads to destruction of the overall alveolar architecture, strong impairment of lung function and eventually death of the patient [216,219].

Nucleotides like ATP and UTP are released from injured epithelial cells [28,133,134,136] and ATP levels are significantly increased in bronchoalveolar lavage (BAL) fluid from IPF patients [117,134]. In line, ATP was increased in a murine bleomycin-induced pulmonary fibrosis model [117,134].

It was also found that P2Y_2_ receptor expression was up-regulated on BAL fluid macrophages and blood neutrophils derived from IPF patients [117], whereas P2Y_6_ receptor expression was upregulated on lung structural cells in the alveolar space of IPF patients but not on BAL cells [220]. Both, P2Y_2_ and P2Y_6_ receptor expression were upregulated in the lung tissue of animals with bleomycin-induced pulmonary fibrosis and activation of P2Y_2_ or P2Y_6_ receptor was associated with enhanced proliferation of human and murine lung fibroblasts [117,220] (Figure 3). P2Y_2_ receptor activation also stimulated the migration of primary lung fibroblasts [117]. Moreover, ATP increased the numbers of neutrophils and macrophages in the lungs of *WT* animals whereas no changes were seen in P2Y_2_-deficient mice [117]. It has recently been shown that monocyte-derived macrophages are recruited to the lung and that selective inhibition thereof ameliorated lung fibrosis [24]. P2Y_2_- as well as P2Y_6_-deficient animals were found to be partially protected from bleomycin-induced pulmonary inflammation and fibrosis [117].

However, data from animal models have to be taken with caution. Moreover, animal models of lung fibrosis do not recapitulate IPF [217]. The widely used bleomycin model triggers an early inflammatory response followed by fibrotic remodeling at a later stage [221], yet IPF does appear to follow a pathogenic sequence of secondary and modulatory immune activation [222]. Hence, it is difficult to dissect the impact of the P2 receptor expression/activation on inflammation or fibrotic changes; moreover, P2Y_2_ as well as P2Y_6_ receptor activation, have been associated with pulmonary inflammation [117,220]. 

### 3.4. Pulmonary Arterial Hypertension (PAH) 

Patients with PAH suffer increased vascular resistance and high pulmonary arterial pressure. In PAH smooth muscle cells (SMC) proliferate in small peripheral pulmonary arteries and ECs form plexiform lesions within the lung tissue [223].

In lungs of patients with idiopathic PAH (IPAH), a decreased expression of CD39 by ECs in plexiform lesions and primary isolated ECs was detected [38,91]. Decreased expression of CD39 leads to an increased ATP level, which could also be measured in the plasma of patients with IPAH [90]. Downregulation of CD39 expression was observed in lung tissue, as well as in primary isolated EC from IPAH patients [38,90]. On the other hand, circulating EC micro particles from IPAH patients have an increased expression of CD39 [224].

Despite CD39 downregulation, *in-vitro* studies of isolated EC from IPAH patients also showed an upregulated P2Y_11_ receptor expression on protein, but not mRNA level [38]. These changes in the purinergic system lead to an apoptosis resistant phenotype of EC, which could be reversed by downregulation of the P2Y_11_ receptor using siRNA. Furthermore, it was shown that the attenuated CD39 expression and increased ATP level increases SMC migration and proliferation, promoting vascular remodeling [38].

On the basis of the downregulation of CD39, a mouse model for IPAH was established where CD39^-/-^ mice were housed under hypoxic conditions [90]. In this model, as well as in tissue samples from IPAH patients, an increased P2X_1_ receptor expression was found [90]. Contribution of the P2X_1_ receptor to IPAH could be demonstrated by rescue experiments with NF279, a P2X_1_ receptor antagonist [90]. NF279 was able to prevent elevation of pulmonary arterial pressure [90].

Further, for PAH due to BMPR2 (Bone morphogenetic protein receptor type II, serine/threonine receptor kinase) loss-of-function mutations or downregulation in pulmonary ECs, RNA sequencing and Ca^2+^ imaging data showed a link to the P2 receptor Ca^2+^-signalosome [32].

## 4. Conclusion and Outlook

It is well established that P2 receptors are widely expressed on the cells in the lung. Extracellular ATP and P2 receptor-dependent signaling is central to fundamental mechanisms maintaining alveolar homeostasis, in particular surfactant secretion and alveolar host defense. It is not surprising therefore that, in recent years, the contribution of P2 receptors and P2-mediated signaling has gained widespread attention for the pathophysiology of alveolar diseases. A better understanding thereof can ultimately culminate in the development of targeted therapeutics. For example, it has been shown that P2X_7_ receptor antagonists reduced neutrophil infiltration and proinflammatory cytokine levels [98]. Various P2X_7_ receptor antagonists are currently in development for the clinic [173]. Also, other P2 receptor agonists and antagonists have already been approved as therapeutics for various diseases, some of which may well be relevant for the treatment of lung diseases in the future. Diquafosol, a long-acting P2Y_2_ receptor agonist has been approved for the treatment of dry eye disease [225]. P2Y_12_ receptor antagonists (clopidogrel, prasugrel, cangrelor and ticagrelor) have become an important class of antithrombotic drugs blocking P2Y_12_ receptor-mediated platelet aggregation [196,226,227]. Successful clinical trials have been completed for application of the P2X_3_ receptor antagonist gefapixant in chronic cough and other inflammatory conditions [196,228].

Apart from the development of specific drugs, a detailed understanding of the pathophysiology is required to bring P2 receptor therapies to the distal lung. Much of the current knowledge has been derived through use of animal models, which offer great possibilities to investigate the role of specific P2 receptors using adequate *knock-out* models. However, caution is required when translating findings from animal studies to the human lung. Animal models for pulmonary disease do not recapitulate the full spectrum of human pathophysiology. This is a result of differences in anatomy, as well as physiology [221,229,230]. It will be important to assess the implication of P2 receptor signaling in human lungs or lung tissue from healthy and diseased donors, but availability and access to such tissue often pose a major limitation. *In-vitro* models offer alternatives to increase our understanding of how the human lung maintains homeostasis and how dysregulation of specific cellular processes leads to disease, especially in hard-to-study lung regions like the human alveolus. Such models can provide powerful, scalable screening platforms to test pharmaceuticals, and can act as an important preclinical step that bridges the gap between drug testing in rodent models and human clinical trials. Recently developed alveolar “lung-on-a-chip” systems recapitulate structural, functional and mechanical elements of the unique biophysical and cellular architecture of the human alveolus *in vitro* [231,232,233,234,235]. Leveraging these *in-vitro* models will certainly help in deciphering the molecular and cellular mechanisms driving pathophysiological alterations and thereby accelerate the discovery of novel therapeutic targets [230].

## Figures and Tables

**Figure 1 ijms-21-04973-f001:**
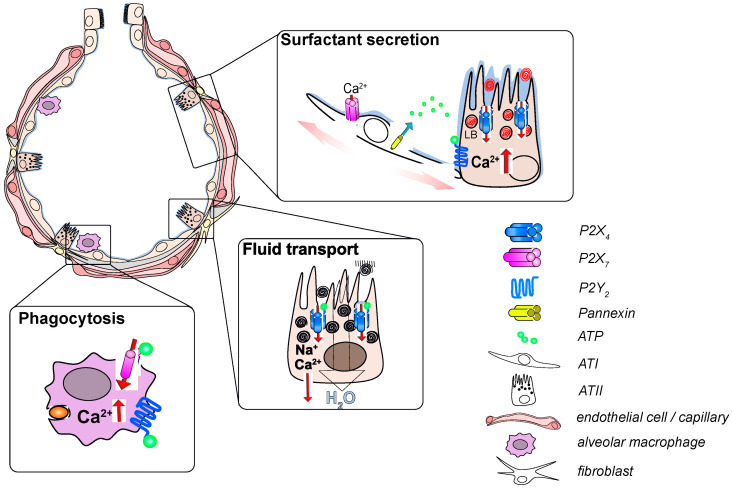
Functional relevance of P2 receptor signaling in the alveolus. Inflation of the alveolus leads to stretch-induced release of ATP (adenosine triphosphate) from alveolar epithelial cells which activates P2Y_2_ receptors on ATII (type II) cells. The resulting Ca^2+^ release from the endoplasmatic reticulum stimulates LB exocytosis. Subsequent activation of P2X_4_ receptors on the limiting membrane of fused LBs results in a fusion-activated Ca^2+^-entry (FACE) which facilitates release of surfactant from fused LBs. FACE also results in transepithelial cation transport leading to fluid resorption from the alveolar lumen to promote activation of secreted surfactant. Activation of P2X_7_ and P2Y_2_ receptors on AMs results in an increase in intracellular Ca^2+^ that facilitates phagocytosis of airborne particulates.

**Figure 2 ijms-21-04973-f002:**
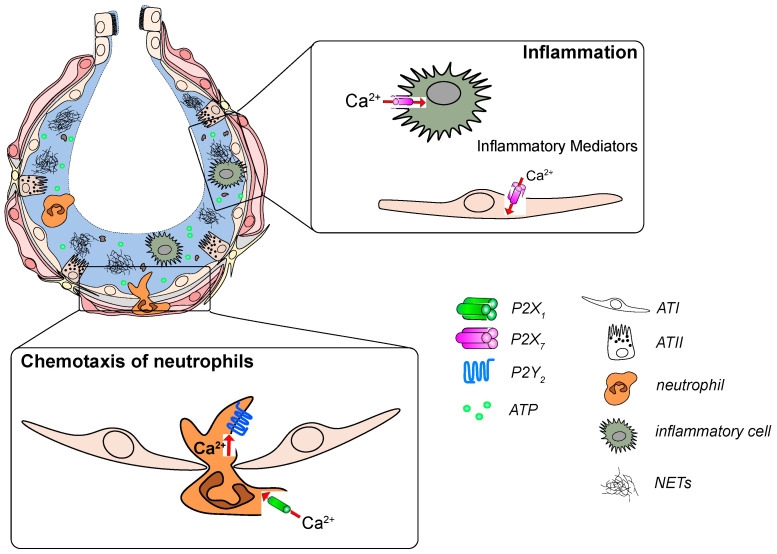
P2 receptor signaling in development and progression of acute respiratory distress syndrome (ARDS)**.** Increased ATP levels act as a “danger signal” in the damaged alveolus. ATP activates P2X_7_ receptors on immune and epithelial cells to promote the release of inflammatory mediators. ATP is also sensed by P2Y_2_ and P2X_1_ receptors on neutrophils which leads to their infiltration into the alveolus and subsequent activation.

**Figure 3 ijms-21-04973-f003:**
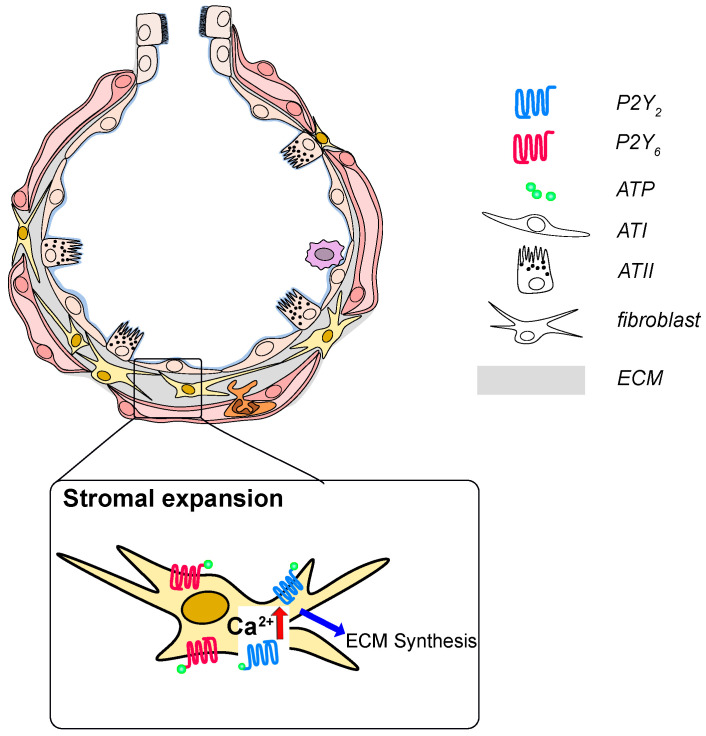
P2 receptor signaling in development and progression of idiopathic pulmonary fibrosis (IPF) ATP and UTP are released from injured epithelial cells and are increase in the BAL fluid from IPF patients. Activation of P2Y_2_ or P2Y_6_ on fibroblasts results in fibroblast proliferation, migration and likely excessive deposition of extracellular matrix constituents.

**Table 1 ijms-21-04973-t001:** Expression of P2 receptors within the cells of the distal lung.

Cell Type	P2 Receptor	Expression
**ATI**	P2X_7_	mRNA (r [27]), protein (r [27])
	P2Y_2_	protein (r [28])
**ATI like**	P2Y_2_	protein (h [28], r [28])
**ATII**	P2X_4_	mRNA (r [29,30]), protein (r [29,30])
	(P2X_7_)	mRNA (r [30]), protein (r [30])
	P2Y_2_	mRNA (r [30,31]), protein (r [30])
	P2Y_6_	mRNA (r [30]), protein (r [30])
**EC**	P2X_1_	mRNA (h [32]), protein (h [32])
	P2X_4_	mRNA (h [32,33,34,35]), protein (h [32])
	P2X_5_	mRNA (h [32,34,35]), protein (h [32])
	P2X_6_	mRNA (h [35])
	P2X_7_	mRNA (h [35])
	P2Y_1_	mRNA (h [32,35,36,37]), protein (h [32,36])
	P2Y_2_	mRNA (h [35,37])
	P2Y_6_	mRNA (h [32]), protein (h [32])
	P2Y_11_	mRNA (h [32,36,37,38]), protein (h [32,36])
	P2Y_12_	mRNA (h [37])
	P2Y_14_	mRNA (h [37])
**Fibroblasts**	P2Y_2_	mRNA (h [39])
	P2Y_4_	mRNA (h [39])
	P2Y_6_	mRNA (h [39])
	P2Y_11_	mRNA (h [39])
**Alveolar macrophages**	P2X_1_	mRNA (h [40], m [41], r [42])
	P2X_3_	mRNA (m [41] h [40])
	P2X_4_	mRNA (h [40], m [41], r [42]), protein (h [43], r [43,44]),
	P2X_5_	mRNA (h [40])
	P2X_7_	mRNA (h [40], r [42])
	P2Y_1_	mRNA (h [40], r [42])
	P2Y_2_	mRNA (h [40], r [42])
	P2Y_4_	mRNA (h [40], r [42])
	P2Y_6_	mRNA (h [40])
	P2Y_11_	mRNA (h [40])
	P2Y_12_	mRNA (r [42])
	P2Y_13_	mRNA (h [40])
	P2Y_14_	mRNA (h [40])
**Neutrophils**	P2X_1_	mRNA (h [45,46,47], r [48]), protein (h [47])
	P2X_4_	mRNA (h [46,47], r [48])
	P2X_5_	mRNA (h [45,46,47], r [48])
	P2X_6_	mRNA (h [47])
	P2X_7_	ambiguous data ([45,46,48,49,50,51,52])
	P2Y_1_	mRNA (h [46])
	P2Y_2_	mRNA (h [46,51,53]), protein (h [51], r [48])
	P2Y_4_	mRNA (h [51])
	P2Y_6_	mRNA (h [51])
	P2Y_11_	mRNA (h [51]), protein (h [45], r [48])
	P2Y_14_	mRNA (h [54])

The table summarizes the evidence for P2 receptor expression in the distal lung that has been reported at the mRNA and protein level in human (h), mouse (m) and rat (r).

**Table 2 ijms-21-04973-t002:** Contribution of P2 receptor signaling to alveolar function and to the development/progression of lung diseases.

P2 Receptor	Physiological Function	Implication in Lung Diseases
**P2X_1_**	Host defense - neutrophil degranulation - enhanced phagocytosis [88] - T cell activation [82]	Transfusion-related acute lung injury [89] Contribution to development of IPAH [90]
**P2X_4_**	Surfactant secretion/ release [29,91,92] Fluid resorption [93] Host defense - T cell activation [82] and migration [94]	Development of VILI [95]
**P2X_7_**	Host defense - Bacterial clearance [52,77,96,97] - Neutrophil infiltration [98] - T cell activation [82] - Inflammasome activation [99,100] - Fast migration of DCs [87] - Sensing of cell damage [101] - Production of type 1 interferons [102] - Pro-apoptotic effect in Tregs [87] - Pyroptotic cell death of phagocytes [103] - Control of iNKT population size [104]	Pulmonary tuberculosis [77] Pathogenesis and progression of ARDS [99,105,106,107,108,109,110,111]
**P2Y_2_**	Surfactant secretion [12,112] Fluid transport [113,114] Host defense - Sensing of apoptotic cells [115] - Chemotaxis of myeloid cells [53,116]	Development of VILI [96] Fibrotic remodelling [117]

The table summarizes the functionally most relevant P2 receptors within the distal lung.

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
