# Peer review of "P2 Purinergic Signaling in the Distal Lung in Health and Disease"

_ijms, 2020, doi:10.3390/ijms21144973_

Round 1
Reviewer 1 Report
In this review, an extensive description of P2 receptors expressed within the cells in the distal lung is presented. In final section, the role of P2 receptors is some lung diseases is included.
- A graphical abstract would be welcome.
- In section 4 “The role of P2 receptors in lung disease”, it would be good to summarize information by including a table with most important P2 receptors involved in the pathological diseases presented in this section.
- References must be numbered in order of appearance in the text (according to guidelines)
Author Response
We would like to thank the reviewer for the useful and constructive criticisms. We have done our best to address all of the0s comments in full. This has certainly helped to improve the manuscript. The reviewer´s comments follow in verbatim, with our responses interspersed red.
In this review, an extensive description of P2 receptors expressed within the cells in the distal lung is presented. In final section, the role of P2 receptors is some lung diseases is included.
- A graphical abstract would be welcome.
We have added a graphical abstract
- In section 4 “The role of P2 receptors in lung disease”, it would be good to summarize information by including a table with most important P2 receptors involved in the pathological diseases presented in this section.
We have now added a second table (table 2) to highlight the most important P2 receptors involved in maintaining homeostasis and the selected pathologies.
- References must be numbered in order of appearance in the text (according to guidelines)
Done
Reviewer 2 Report
Wirsching et al. have written a comprehensive and well-written review on the literature surrounding the role of purinergic receptors in the lung, including their contribution to alveoli form and function in health and disease from human and rodent studies.
I understand that this review forms part of a special issue on purinergic receptors, however, I feel that it is still important to provide more introduction and background information on these receptors, for those who read this article alone. Specifically, what is the difference in physiology and function of the different P2 family members? The authors mention that many cell types in the lung express purinergic receptors, is the lung particularly enriched in these receptors compared to other tissues? If so, why?
Related to the above point, the first section on expression levels on different cell types provides basic information, however, I feel that it could be enhanced with more discussion on the significance of different expression profiles on different cell types.
As a non-specialist, there were a lot of terms that were not familiar, e.g. patch-clamp, hypophase – could the authors provide a glossary? Exactly what the surface molecules CD39 and CD73 are, is not elaborated on.
After reading the review, I have a number of questions that the authors could also endeavor to address. The authors highlight that both immune cells and ATII cells release ATP, what are the stimuli that drive ATP release? What is the function, or outcome, of ATP release?
Further, ATP as a DAMP can activate production of alarmins, including IL-33 (Kouzaki et al. 2011), suggesting downstream pathways are important for lung immunity. This view is further supported by the fact that NLRP3 is also activated by P2 receptors, which is touched on by the authors. Could they elaborate further on the contribution of ATP and P2 receptors specifically in immune activation? Given the role of ATP and P2 in driving alarmin production, what is the significance of these effectors in allergy and asthma?
Finally, the authors mention a number of issues that could be discussed a little more extensively. What are the specific reasons for gaps in knowledge? Why is it that rodent models do not recapitulate human physiology in this context? What are the relevant in vitro models that could be used in place of rodents? Are there any therapeutics in clinic or trial that target P2 receptors, or that are used in other spheres that could be relevant to lung disease?
Author Response
We would like to thank the reviewer for the useful and constructive criticisms. We have done our best to address all of the comments in full. This has certainly helped to improve the manuscript. The reviewer´s comments follow in verbatim, with our responses interspersed red.
Wirsching et al. have written a comprehensive and well-written review on the literature surrounding the role of purinergic receptors in the lung, including their contribution to alveoli form and function in health and disease from human and rodent studies.
We very much appreciate the positive evaluation of our review by the referee
I understand that this review forms part of a special issue on purinergic receptors, however, I feel that it is still important to provide more introduction and background information on these receptors, for those who read this article alone. Specifically, what is the difference in physiology and function of the different P2 family members? The authors mention that many cell types in the lung express purinergic receptors, is the lung particularly enriched in these receptors compared to other tissues? If so, why?
We have now included additional paragraphs providing background to P2 receptor nomenclature, pharmacology and cellular function.
Related to the above point, the first section on expression levels on different cell types provides basic information, however, I feel that it could be enhanced with more discussion on the significance of different expression profiles on different cell types.
Following the suggestion of the referee and also in line with the suggestions from reviewers 1 and 3, we have now reorganized the section on expression and physiological function, more clearly highlighting what we believe are the most relevant P2 receptors in health and disease. We have also added an additional table (table 2) to highlight these P2 receptors.
As a non-specialist, there were a lot of terms that were not familiar, e.g. patch-clamp, hypophase – could the authors provide a glossary? Exactly what the surface molecules CD39 and CD73 are, is not elaborated on.
We have added a glossary at the beginning
After reading the review, I have a number of questions that the authors could also endeavor to address. The authors highlight that both immune cells and ATII cells release ATP, what are the stimuli that drive ATP release? What is the function, or outcome, of ATP release?
We have extended the section on ATP in the alveolus adding some more detail regarding release of ATP under pathophysiological conditions. Details were recently reviewed within this Journal – this has been highlighted now. we have also highlighted that under “normal” conditions extracellular ATP levels are very low in the alveolar lumen and that release (either physiologic or in response to damage/inflammation) is necessary for P2 receptor activation.
Further, ATP as a DAMP can activate production of alarmins, including IL-33 (Kouzaki et al. 2011), suggesting downstream pathways are important for lung immunity. This view is further supported by the fact that NLRP3 is also activated by P2 receptors, which is touched on by the authors. Could they elaborate further on the contribution of ATP and P2 receptors specifically in immune activation? Given the role of ATP and P2 in driving alarmin production, what is the significance of these effectors in allergy and asthma?
We have added the indicated reference and included the information regarding ATP – P2 receptor signaling and NLRP3 activation. As outlined by the reviewer, this has been mainly investigated in the context of airway diseases, in particular allergic asthma – which is not a focus of this review on the distal lung.
Finally, the authors mention a number of issues that could be discussed a little more extensively. What are the specific reasons for gaps in knowledge? Why is it that rodent models do not recapitulate human physiology in this context? What are the relevant in vitro models that could be used in place of rodents? Are there any therapeutics in clinic or trial that target P2 receptors, or that are used in other spheres that could be relevant to lung disease?
We have extended the Conclusion and outlook session to address the points raised
Reviewer 3 Report
1. There is a large mount of excellent material here, but I suggest that it needs to be organised and illustrated in a much more cohesive, interesting and "delight to read" format. Much at the moment is tough to get though and frankly in places rather boring. The authors are not capitalising on the thorough tables they have.
2.So, could I suggest that Section 1 becomes a description of the system, its receptors and ligands, and the cellular functional consequences of receptor activation by different ligands and receptor polymerisation etc. The various numeric suffices for receptors are not explained. Thus, more of an overarching explanatory approach to the basic pharmacology.
3. This would then role into Section 2 on the cells and tissue elements served by the purinergic system, but no point in just repeating the information in the table. Use the individual figures, now bunched together, to integrate with and illustrate individual parts of the text on the mechanistic features of the system in its various parts. Try and emphasise what is likely to be most important and most worked out. What are the chief messages you want the reader to take away?
4. The section of actual diseases was pretty good I thought, and needs little change.
5. Conclusions: I`m not sure that new animal models are the answer to the main issue that is raised here; instead my prejudice is more use of clinical material in that the best model of the disease is the patient and their lungs, especially if you can get your hands on some of it.
Lesser points:
5. Syntax: There are some odd use of words, but I appreciate the trans-linguistic nature of the work; and overall the English is good. For example, eminent for a disease is not quite right; "important" or "major" would be better, with some criterion for that assessment. There are a number of examples like that which they or the editor could search for. The occasion sentence has no verb; receptors should be "on" cells generally and not "in", unless that is deliberate when it needs explanation. Some verbs should be singular/plural according to he exact noun they are accompanying. Compound words are best with a hyphen eg activity-amplification.
6. Some if the technical words (eg, hypophase) and abbreviations (eg ASL) need definition as soon as used, especially if already referred to using a different term. I`m not sure I like the term "distal lung": I divide the respiratory system into airways and lungs, which is what you mean (when I read "distal lung" I think it means sub-pleural alveolae. You also use the term "septum" for what I would call the "respiratory membrane"; I use the term septum/ae to imply connective tissue support structure(s)
Author Response
We would like to thank the reviewer for the useful and constructive criticisms. We have done our best to address all of the comments in full. This has certainly helped to improve the manuscript. The reviewer´s comments follow in verbatim, with our responses interspersed red.
There is a large mount of excellent material here, but I suggest that it needs to be organised and illustrated in a much more cohesive, interesting and "delight to read" format. Much at the moment is tough to get though and frankly in places rather boring. The authors are not capitalising on the thorough tables they have.
- So, could I suggest that Section 1 becomes a description of the system, its receptors and ligands, and the cellular functional consequences of receptor activation by different ligands and receptor polymerisation etc. The various numeric suffices for receptors are not explained. Thus, more of an overarching explanatory approach to the basic pharmacology.
We have now included additional paragraphs providing background to P2 receptor nomenclature, pharmacology and cellular function.
- This would then role into Section 2 on the cells and tissue elements served by the purinergic system, but no point in just repeating the information in the table. Use the individual figures, now bunched together, to integrate with and illustrate individual parts of the text on the mechanistic features of the system in its various parts. Try and emphasise what is likely to be most important and most worked out. What are the chief messages you want the reader to take away?
Following the reviewer´s suggestion, we have now shortened the section summarizing expression and attempted to more specificially highlight what we believe are the most relevant P2 receptors. We have also added an additional table (table 2) to highlight these P2 receptors.
- The section of actual diseases was pretty good I thought, and needs little change.
Appreciated
- Conclusions: I`m not sure that new animal models are the answer to the main issue that is raised here; instead my prejudice is more use of clinical material in that the best model of the disease is the patient and their lungs, especially if you can get your hands on some of it.
We have added this view to the final section and also elaborated a little more, why we believe that representative, humanized in vitro systems are valuable tools to better understand the contribution of P2 receptor function to maintaining alveolar homeostasis and in the development and/or progression of lung diseases.
Lesser points:
- Syntax: There are some odd use of words, but I appreciate the trans-linguistic nature of the work; and overall the English is good. For example, eminent for a disease is not quite right; "important" or "major" would be better, with some criterion for that assessment. There are a number of examples like that which they or the editor could search for. The occasion sentence has no verb; receptors should be "on" cells generally and not "in", unless that is deliberate when it needs explanation. Some verbs should be singular/plural according to he exact noun they are accompanying. Compound words are best with a hyphen eg activity-amplification.
We have tried our best to correct “odd use of words” and address grammatical errors.
- Some if the technical words (eg, hypophase) and abbreviations (eg ASL) need definition as soon as used, especially if already referred to using a different term. I`m not sure I like the term "distal lung": I divide the respiratory system into airways and lungs, which is what you mean (when I read "distal lung" I think it means sub-pleural alveolae. You also use the term "septum" for what I would call the "respiratory membrane"; I use the term septum/ae to imply connective tissue support structure(s)
We added definitions at the first use of technical words and, following a suggestion from reviewer 2, also added a glossary at the beginning to make the manuscript a easier read overall. With regards to the term “distal lung” we agree that a division into airways and lungs is often used. We have in most places deleted “distal” or if appropriate replaced “distal lung” it with “alveolus”. However, in some instances, the term lung can also be misleading (i.e. including airways and parenchyma) and we therefore maintained “distal lung” in selected cases where we want to focus on the anatomic structure of the lung parenchyma without airways. Regarding “alveolar septum”, this is again a well-established term and defines the entire structure of the alveolar wall (including ECM and structural cells), whereas respiratory membrane often is limited to the thin structure formed of epithelial and endothelial cells relevant for gas exchange. Again, in places where we feel that we need to include all the cells and components in the alveolar structures we use the term alveolar septum.
Round 2
Reviewer 3 Report
I am very pleased at the way the authors have restructured the manuscript to make it both much more readable and also more informative. It is now a very exciting article for a general and respiratory audience.
One major stylistic feature that should be improved is to subdivide the oversized lumps of text into sub-paragraphs. This could be easily done by making logical divisions as new ideas emerge or current ones diverge. Again, it is about attractive readability.
There are places where abbreviations need spelling out in full, even if already done, e.g with LBs and ASL. It is distracting to go off to try to search for where this was first used, when it becomes central feature. This is a place here a bit of repetition would be useful.
Fig 3 legend: there needs to be a stop between ARDS and ATP.
Need to make sure that all new sentences have verbs (rather than just adverbs)!
I would now recommend acceptance with these relatively minor changes.
Author Response
We would like to thank the reviewer for the useful and constructive criticisms. We have done our best to address all of the comments in full. This has certainly helped to improve the manuscript. The reviewer´s comments follow in verbatim, with our responses interspersed red.
I am very pleased at the way the authors have restructured the manuscript to make it both much more readable and also more informative. It is now a very exciting article for a general and respiratory audience.
We appreciate that the reviewer approves our changes and thank for the positive appraisal of the manuscript.
One major stylistic feature that should be improved is to subdivide the oversized lumps of text into sub-paragraphs. This could be easily done by making logical divisions as new ideas emerge or current ones diverge. Again, it is about attractive readability.
We have inserted breaks to subdivide the text
There are places where abbreviations need spelling out in full, even if already done, e.g with LBs and ASL. It is distracting to go off to try to search for where this was first used, when it becomes central feature. This is a place here a bit of repetition would be useful.
We have inserted spelled out terms as suggested
Fig 3 legend: there needs to be a stop between ARDS and ATP.
Done
Need to make sure that all new sentences have verbs (rather than just adverbs)!
We have checked the manuscript thoroughly, inserted verbs where missing and corrected grammar and spelling mistakes where necessary
I would now recommend acceptance with these relatively minor changes.